# VIDEO-ToC: VIDEO TREE-OF-CUE REASONING

## ABSTRACT

Existing Video Large Language Models (Video LLMs) struggle with complex video understanding, exhibiting limited reasoning capabilities and potential hallucinations. In particular, these methods tend to perform reasoning solely relying on the pretrained inherent reasoning rationales whilst lacking perception-aware adaptation to the input video content. To address this, we propose **Video-ToC**, a novel video reasoning framework that enhances video understanding through tree-of-cue reasoning. Specifically, our approach introduces three key innovations: (1) A tree-guided visual cue localization mechanism, which endows the model with enhanced fine-grained perceptual capabilities through structured reasoning patterns; (2) A reasoning-demand reward mechanism, which dynamically adjusts the reward value for reinforcement learning (RL) based on the estimation of reasoning demands, enabling on-demand incentives for more effective reasoning strategies; and (3) An automated annotation pipeline that constructs the Video-ToC-SFT-1k and Video-ToC-RL-2k datasets for supervised fine-tuning (SFT) and RL training, respectively. Extensive evaluations on six video understanding benchmarks and a video hallucination benchmark demonstrate the superiority of Video-ToC over baselines and recent methods. All code, models, and datasets will be released.

## 1 INTRODUCTION

Video Large Language Models (Video LLMs) have achieved significant progress on various perception-based video understanding tasks Zhang et al. (2024c); Bai et al. (2025); Wang et al. (2024b). Despite their strong performance on these benchmarks, they often lack reasoning capability and struggle with complex video reasoning tasks.

Recently, inspired by the success of DeepSeek-R1 Guo et al. (2025), which introduces Reinforcement Learning (RL) to greatly improve the model's reasoning abilites in text-based domains, many efforts Feng et al. (2025); Li et al. (2025a) explore applying RL to Video-LLMs for enhancing video reasoning. The common practice to train such a video reasoning model usually includes two stages. First, the supervised fine-tuning (SFT) on video QA samples with labeled reasoning process is performed to cold start the model for adapting the reasoning-based answering style. The followed RL stage further incentivize the model to explore more effective and general reasoning strategies.

The labeled rationales in the training samples of SFT cold start stage is crucial, which basically determines the reasoning style of the model. However, current methods Feng et al. (2025) usually leverage strong models (e.g. Qwen2.5-VL-72B Bai et al. (2025)) to freely generate these rationales without a tailored reasoning pattern, which is not suitable for much smaller models (e.g. Qwen2.5-VL-7B Bai et al. (2025)) to learn and imitate. This is because the smaller model owns relatively weaker spatio-temporal perception capability, which hinders effective reasoning when the model cannot capture enough useful visual cues from the video. Therefore, some reasoning strategies inherent in these rationales encourage the model to rely more on prior language knowledge rather than the provided video semantics, which increases the risk of hallucination Li et al. (2025b). As shown in Figure 1 for example, when the solution of a question requires fine-grained visual cues, Video-R1 Feng et al. (2025) will easily forget searching for key information in the video and start analyzing the question totally based on its prior language knowledge. This observation naturally leads to our core research question: *Can we develop a progressive visual cue localization approach to enhance perception capabilities and mitigate hallucination?*

**Our solution.** To tackle this challenge and improve the model's reasoning strategies, we develop a reasoning framework called 'Video-ToC', which is based on tree-guided visual cue localization.

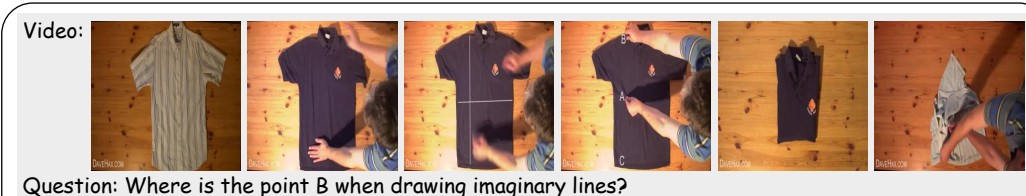

Video:

Question: Where is the point B when drawing imaginary lines?
A. On the center of the shirt. B. On the bottom of the shirt. C. On the shoulder.

Video-R1: <think>Let me think about this for a moment. The question is asking about......The video shows......Now, let's consider the options: A. This seems plausible because the center of the shirt is a common reference point for drawing lines. B. This is less likely because the bottom of the shirt is not typically used as a reference point for drawing lines. C. This is also less likely because the shoulder is not a common reference point for drawing lines on a shirt. Hmm, it seems that the most logical and common reference point for drawing lines on a shirt is the center......Oh, I see. The point B is indeed on the center of the shirt......</think>\n<answer>A</answer>

Video-ToC: <locate>I start by scanning the video to identify any scenes that involve drawing lines on a shirt......Next, I look for a scene where a person is drawing lines on a shirt......Then, I pay close attention to the placement of the lines and any points that are labeled......Finally, I carefully observe the points that are labeled, specifically looking for the point B to determine its location on the shirt......</locate>\n<answer>C</answer>

Figure 1: Reasoning strategy comparison between Video-R1 and our Video-ToC.

An example of our Video-ToC rationale is shown in Figure 1, which demonstrates the process of progressively locating key spatio-temporal visual cues that become increasingly helpful for answering the question. This rationale, characterized by step-by-step localization, enables the model to meticulously examine fine-grained details within the video during question analysis, which is beneficial for mitigating hallucination and handling tasks that require precise perceptual capabilities.

To facilitate learning of this reasoning process, we construct the 'Video-ToC-SFT-1k' dataset for supervised fine-tuning (SFT). The dataset is built upon a tree-based data structure representing video clips, where each leaf node corresponds to the content of an individual clip. The reasoning localization trajectory is derived by traversing paths from the root of the tree to critical leaf nodes, followed by summarization via a large language model (LLM). Then, in the following RL stage, we employ GRPO Shao et al. (2024b) and introduce Reasoning Demand—a metric quantifying the question's reasoning complexity, computed as the error rate when the model answers without reasoning over multiple trials. We further propose Reasoning-demand Reward, proportional to this demand, as the success reward. Unlike GRPO's binary reward, our design better incentivizes useful reasoning strategies, enhancing the model's reasoning ability. Using this framework, we construct the 'Video-ToC-RL-2k' dataset with reasoning demand annotations for GRPO training.

Equipped with Video-ToC, the model achieves robust reasoning capabilities when handling queries that demand intricate spatio-temporal perception. It outperforms other reinforcement learning-based methods on a series of challenging video understanding and video hallucination benchmarks, including VSI-Bench Yang et al. (2024), VideoMMMU Hu et al. (2025), MMVU Zhao et al. (2025), MVBench Li et al. (2024c), TempCompass Liu et al. (2024b), VideoMME Fu et al. (2024), and VideoHallucer Wang et al. (2024c), demonstrating its clear advantage.

To summarize, we make the following contributions:

- We present Video-ToC, which is a novel video reasoning framework that introduces a tree-guided visual cue localization mechanism and a reasoning-demand-based reward strategy. This approach endows the model with enhanced fine-grained perceptual capabilities through structured reasoning patterns.

- For acquiring the fine-grained reasoning ability, we develop an automatic data generation pipeline to construct two video reasoning datasets, i.e., Video-ToC-SFT-1k and Video-ToC-RL-2k, for SFT and RL training, respectively.

- Comprehensive evaluations across six video understanding benchmarks and one video hallucination benchmark substantiate the efficacy of our method, demonstrating consistent performance improvements and hallucination mitigation.

## 2 RELATED WORK

### 2.1 VIDEO LARGE LANGUAGE MODELS

As a kind of Multimodal Large Language Models (MLLMs) Liu et al. (2023); Zhu et al. (2023); Zhang et al. (2024a); Li et al. (2024a); Bai et al. (2025) especially designed for video data, Video Large Language Models (Video LLMs) Maaz et al. (2024); Zhang et al. (2023a); Liu et al. (2024a); Li et al. (2024d); Wang et al. (2024b); Zhang et al. (2024c) have shown remarkable capabilities in comprehending and analyzing complex spatio-temporal visual cues within videos. For example, VideoChatGPT Maaz et al. (2024) disentangles the spatial and temporal features in a dual-pathway framework, enabling efficient video features modeling. Video-LLaMA Zhang et al. (2023a) employs the Q-Former Li et al. (2023) for feature compression and introduces an audio branch to integrate more diverse multimodal information. ST-LLM Liu et al. (2024a) delegates the task of video sequence modeling to the LLMs through the proposed dynamic masking strategy with specifically designed training objectives. Despite these advancements significantly enhancing the perception abilities of Video LLMs, their reasoning capabilities still remain underexplored Feng et al. (2025); Li et al. (2025a); Zhang et al. (2025b).

### 2.2 MULTIMODAL LARGE LANGUAGE MODEL REASONING

Recent studies focusing on the reasoning abilities of MLLMs highlight the great potential of tackling complex tasks through Chain-of-Thought (CoT) reasoning Wei et al. (2022); Zhang et al. (2023b). The general paradigm to improve the reasoning capabilities of MLLMs is performing supervised fine-tuning (SFT) using a collection of high-quality CoT reasoning data annotated by powerful models (e.g., GPT-4) and/or humans Wu & Xie (2024); Han et al. (2024); Fei et al. (2024); Shao et al. (2024a); Qi et al. (2024); Xu et al. (2024). However, merely teaching the models to memorize thinking-style reasoning paths leads to limited generalizability Chu et al. (2025), which can be greatly alleviated by reinforcement learning which incentivizes the reasoning capabilities in MLLMs Zhang et al. (2025a); Liu et al. (2025); Wang et al. (2025). While this approach remarkably improves performance on strong reasoning data which are math-related Lu et al. (2023); Wang et al. (2024a) or task-specific data like grounding Lai et al. (2024), its effectiveness for video understanding is underexplored. In this work, we aim to enhance the reasoning capabilities of MLLMs and boost their performance on both video reasoning and video general tasks through the development of a high-quality, tailor-made CoT dataset and an improved RL reward design.

## 3 METHOD

### 3.1 OVERVIEW

The training phase of Video-ToC involves two stages: supervised fine-tuning (SFT) and reinforcement learning (RL). In the SFT stage (Sec. 3.2), we detail the rationale annotation pipeline for constructing the training data, while the RL stage (Sec. 3.3) extends beyond the standard accuracy reward by introducing a Reasoning-demand Reward, supported by a dedicated dataset tailored for RL optimization.

### 3.2 DATA CONSTRUCTION FOR SUPERVISED FINE-TUNING (SFT)

The SFT stage of Video-ToC differs from recent approaches Feng et al. (2025) by employing a tree-structured representation of video clips based on their semantic correlations. Each leaf node corresponds to a video clip's content, while the hierarchical structure captures their relationships. To generate SFT data, we simply backtrack from any leaf node to the root, extracting a coherent reasoning path. This rationale is then processed and summarized by an external LLM to produce the final SFT data. The specific steps are detailed as follows.

**Step 1: Leaf node construction.**  To construct a hierarchical tree structure of video clips, we first obtain the leaf nodes by segmenting the input video and extracting their content.

Specifically, as shown in Figure 2, given a sampled video and corresponding question-answer pair, we first segment the video into multiple clips, employing the video splitting method proposed by Panda-70M Chen et al. (2024). In detail, the video is first split based on shot boundary detection,

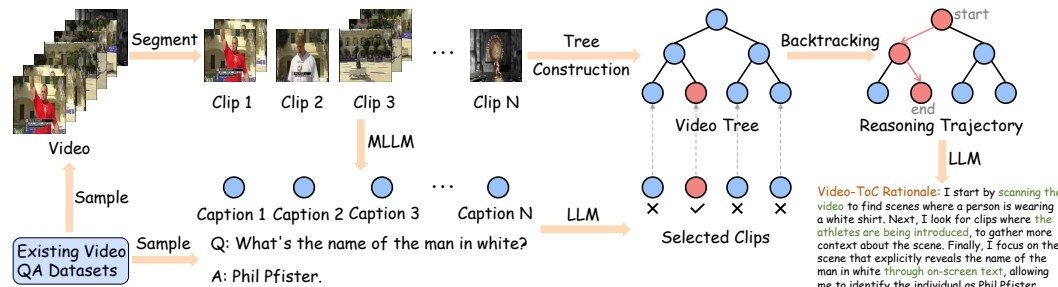

Figure 2: Video-ToC rationale annotation pipeline. The pipeline consists of three phases: (i) **Leaf node construction** through an LLM selecting question-relevant clips, (ii) **Reasoning trajectory generation** through backtracking from the selected leaf nodes to the root node, and (iii) **SFT data construction** through LLM summarization of the reasoning trajectory into the Video-ToC rationale. Details of each phase are presented in Sec. 3.2.

then some adjacent clips are stitched if the frame embeddings from them are similar enough. After segmenting the video into $N$ clips, we prompt an MLLM to describe each clip comprehensively, thereby obtaining $N$ detailed video clip captions. Then we utilize an LLM to analyze the question-answer pair, and identify the key clips that are essential for answering the question based on the provided clip captions. After acquiring the video clips, we construct a Segment Tree with $N$ leaf nodes, where each leaf node represents a distinct video clip.

**Step 2: Reasoning trajectory generation.** Subsequently, by performing backtracking from each selected leaf node (corresponding to a target clip) up to the root, the resulting paths collectively form a subtree that implicitly encodes a reasoning trajectory. This trajectory begins with the entire video as the root, progressively narrows down to finer-grained segments through hierarchical decomposition, and ultimately converges on the key clips, effectively capturing the spatio-temporal localization process in a structured and interpretable manner.

To facilitate comprehension by LLM, the trajectory is preprocessed into multiple visual cue descriptions, where each layer of the subtree is transformed into a 'video compilation' by concatenating all clips associated with its constituent nodes. Formally, for the $i$-th layer of subtree $\mathcal{T}$ containing $k$ nodes $\{\mathcal{T}_{i,j}\}_{j=1}^{k}$, the corresponding compilation $\mathcal{V}_i$ is constructed as

$$\mathcal{V}_i = \text{Concat}(\mathcal{S}(\mathcal{T}_{i,1}), \ldots, \mathcal{S}(\mathcal{T}_{i,k})), \tag{1}$$

where $\mathcal{S}(\mathcal{T}_{i,j})$ denotes the set of leaf nodes rooted at $\mathcal{T}_{i,j}$. To ensure uniqueness, duplicate compilations, resulting from identical clip sets across different layers, are removed, yielding a concise and non-redundant representation of the hierarchical reasoning trajectory. This processed trajectory is then summarized by an LLM as the Video-ToC rationale, effectively bridging the structured decomposition with high-level reasoning. More illustrations are provided in the supplementary material.

**Step 3: SFT data construction.** We first prompt an MLLM to describe the rest 'video compilations' respectively. Each description corresponds to a step in the localization process, detailing the specific spatial and temporal visual cues that the model needs to focus on. Subsequently, we employ an LLM to assess and filter samples where the visual cues from the final step are insufficient to derive the question's answer. Finally, we utilize these visual cue descriptions alongside the question-answer pair to prompt an LLM to generate a natural and coherent narrative serving as the Video-ToC rationale. Such rationales demonstrate the process of locating video clips that are increasingly helpful for solving the question and reaching the answer, as exemplified in the bottom-right portion of Figure 2. More examples are provided in the supplementary material. The Video-ToC rationale, combined with the video and question-answer pair, constitutes a training sample for the SFT cold start stage.

To construct the SFT training data, we apply the above annotation pipeline to the LLaVA-Video-178K dataset Zhang et al. (2024c) by employing Qwen2.5-VL-7B Bai et al. (2025) as the MLLM and Llama-3.3-70B-Instruct Grattafiori et al. (2024) as the LLM. By randomly selecting a small subset of videos and their corresponding question-answer pairs, we curate the Video-ToC-SFT-1k dataset, which comprises 1,000 high-quality training samples designed to facilitate an effective and efficient SFT cold start.

### 3.3 REASONING-DEMAND REWARD FOR REINFORCEMENT LEARNING (RL)

After the SFT cold start stage, we perform reinforcement learning (RL) with the Group Relative Policy Optimization (GRPO) Shao et al. (2024b) algorithm to enable the model to further enhance its reasoning capabilities. The adopted training rewards and objectives are as follows.

**Vanilla accuracy reward.** In GRPO, the vanilla accuracy reward is typically a binary (0-1) function where the value is determined by whether the model's prediction aligns with the question's answer:

$$R_{\text{vanilla}} = \begin{cases} 1, & \text{if } A_{\text{pred}} \text{ is correct} \\ 0, & \text{otherwise,} \end{cases} \tag{2}$$

where $A_{\text{pred}}$ is the predicted answer after thinking. However, solving different questions requires varying degrees of thinking: reasoning-based questions depend more heavily on analytical thought, whereas perception-based ones rely less on it. Therefore, providing the same reward for each correct answer is not the optimal approach to incentivize the effective reasoning strategies for solving reasoning-based questions.

**Reasoning-demand reward.** To tailor a more suitable reward for each training sample, we first assess its reasoning demand and then develop a corresponding reward based on this. Specifically, for a given question, we employ an MLLM to directly predict the answer without thinking in $M$ independent trials and record the number of correct predictions, denoted as $\alpha$ (where $\alpha$ ranges from 0 to $M$). We define the reasoning demand for this question as $e^{-\frac{\alpha}{M}}$, and set the value of reasoning-demand reward equal to it when the model successfully solves the question during training. Formally, the reasoning-demand reward is defined as:

$$R_{\text{rd}} = \begin{cases} e^{-\frac{\alpha}{M}}, & \text{if } A_{\text{pred}} \text{ is correct} \\ 0, & \text{otherwise,} \end{cases} \tag{3}$$

where $A_{\text{pred}}$ is the predicted answer after thinking. The core idea behind Equation equation 3 is that when the model can answer questions accurately without reasoning (large $\alpha$), the need for prior reasoning diminishes (minimize $R_{\text{rd}}$); conversely, poorer model performance (small $\alpha$) requires more extensive reasoning (increase $R_{\text{rd}}$). The exponential function is used to modulate reward magnitudes across different tiers of reasoning demands. Specifically, the reasoning-demand reward escalates rapidly as the accuracy of direct answering declines, while it decreases relatively slowly as successful direct predictions increase.

To this end, the reasoning demand-driven reward mechanism incentivizes the model when a question inherently requires reasoning and the model successfully addresses it through reasoning analysis. Conversely, for perception-based questions requiring minimal reasoning, the reward decreases proportionally. With this reward, the model is guided to make decisions on whether to engage in reasoning, consequently alleviating the problem of overthinking when unnecessarily complex reasoning is applied to straightforward questions.

**GRPO training objective.** During GRPO training, the model first generates a group of $G$ candidate responses $o = \{o_1, \ldots, o_G\}$ for each input question. Then, we calculate the reasoning-demand rewards for each response using Equation equation 3, which serve as their respective final rewards denoted by $\{r_1, \ldots, r_G\}$. Note that the overall reward we apply during GRPO training is only the reasoning-demand reward, and we do not use format reward because the model after the cold start phase can adhere to the specified format well enough. (More discussions are provided in the supplementary material.) Subsequently, GRPO normalizes these rewards as the relative advantages of the responses within a group:

$$A_i = \frac{r_i - \text{mean}(\{r_i\}_{i=1}^{G})}{\text{std}(\{r_i\}_{i=1}^{G})}, \tag{4}$$

where $A_i$ represents the relative advantage of the $i$-th response. Since Equation equation 4 eliminates the reward differences across responses to different questions, we then multiply $A_i$ by the question's reasoning demand (denoted as $\gamma$) to derive the final advantage of the $i$-th response, denoted as $\hat{A}_i$:

$$\hat{A}_i = A_i \times \gamma. \tag{5}$$

Ultimately, the model is optimized through maximizing the following training objective of GRPO:

$$\mathbb{E}_{q,\{o_i\}}\Big[\frac{1}{G}\sum_{i=1}^{G}\Big(\min\big(\frac{\pi_\theta(o_i|q)}{\pi_{\theta_{\text{old}}}(o_i|q)}\,\hat{A}_i,\text{clip}(\frac{\pi_\theta(o_i|q)}{\pi_{\theta_{\text{old}}}(o_i|q)},1-\epsilon,1+\epsilon)\,\hat{A}_i\big)-\beta\,\mathbb{D}_{\text{KL}}(\pi_\theta\,\|\,\pi_{\text{ref}})\Big)\Big],\quad(6)$$

where $\pi_\theta$ and $\pi_{\theta_{\text{old}}}$ represent the current and old policy. $\epsilon$ is a hyperparameter that controls the clipping range. The KL-divergence term $\mathbb{D}_{\text{KL}}(\cdot\|\cdot)$ is introduced to constrain the deviation of $\pi_\theta$ from the reference model $\pi_{\text{ref}}$, with $\beta$ as a hyperparameter controlling the regularization strength.

To construct the RL training data, we only need to annotate the reasoning demand for each sample. This is because RL promotes free exploration by the model, eliminating the need for annotated Video-ToC rationales. The training samples for RL is also derived from a subset of the LLaVA-Video-178K dataset Zhang et al. (2024c). For each video QA, we employ Qwen2.5-VL-7B Bai et al. (2025) to generate answers across 8 independent trials ($M = 8$). We then compute two key metrics: (1) $\alpha$, the count of correct predictions across these trials, and (2) the reasoning demand $e^{-\frac{\alpha}{M}}$, which quantifies each sample's complexity. After balancing samples across different reasoning demand tiers, we construct the final Video-ToC-RL-2k dataset containing 2,000 samples for GRPO training. More details regarding the data construction pipeline are presented in the supplementary material.

## 4 EXPERIMENTS

### 4.1 EXPERIMENTAL SETUP

**Benchmarks.** We evaluate our model on seven widely used video understanding and video hallucination benchmarks, including three video reasoning benchmarks: VSI-Bench Yang et al. (2024), VideoMMMU Hu et al. (2025), and MMVU Zhao et al. (2025), three video general benchmarks: MVBench Li et al. (2024c), TempCompass Liu et al. (2024b), and VideoMME Fu et al. (2024), as well as a video hallucination benchmark VideoHallucer Wang et al. (2024c). Among the video reasoning benchmarks, VSI-Bench focuses on assessing the model's spatial reasoning ability, whereas both VideoMMMU and MMVU primarily evaluate the knowledge acquisition and utilization capabilities. The video general benchmarks contain both reasoning and perception tasks, thus offering a more comprehensive assessment of the model's holistic video understanding abilities. VideoHallucer evaluates hallucination risks on five different task categories, including object-relation, temporal, semantic detail, extrinsic factual, and extrinsic non-factual hallucinations. To be consistent with Video-R1 Feng et al. (2025), we choose the multiple-choice question set for MMVU and evaluate VideoMME without subtitle assistance.

**Implementation details.** Following Video-R1 Feng et al. (2025), we choose Qwen2.5-VL-7B Bai et al. (2025) as the baseline. During the training stage, we first perform supervised fine-tuning (SFT) as the cold-start, on our Video-ToC-SFT-1k dataset for one epoch. The model after the SFT stage is termed as Video-ToC-SFT. Then we conduct reinforcement learning (RL) using GRPO algorithm Shao et al. (2024b) with the proposed reasoning-demand reward, on our Video-ToC-RL-2k dataset for one epoch, to obtain the final Video-ToC model. During both SFT and RL stages, a video is uniformly sampled 16 frames as input and each frame is limited to a resolution of $128 \times 28 \times 28$. For inference, we increase input frame resolution to $256 \times 28 \times 28$.

### 4.2 MAIN RESULTS

We conduct a comprehensive evaluation on Video-ToC's overall video understanding capability and hallucination, comparing it with baseline and recent methods (in particular, Video-R1 Feng et al. (2025), the previous state-of-the-art model), as shown in Table 1 and Table 2.

As shown in Table 1 concerning a series of video understanding benckmarks, in comparison with baseline, our SFT model Video-ToC-SFT significantly boosts performance with only 1,000 training samples. It also largely outperforms Video-R1-SFT, which is the model after the SFT stage of Video-R1, and even performs comparably with Video-R1. This result not only demonstrates the effectiveness of our designed Video-ToC rationales but also emphasizes the importance of teaching the model to locate key visual cues step-by-step during the reasoning process. The reinforcement learning stage leveraging the proposed reasoning-demand reward serves to guide the model beyond the rigid reasoning pattern introduced by supervised fine-tuning, which further enhances performance on the basis of our SFT model. Our final model Video-ToC consistently outperforms all previous methods

Table 1: Accuracy comparison on three video reasoning benchmarks and three video general benchmarks. "Avg." denotes average accuracy of the six benchmarks.

| Method | Frames | Video Reasoning Benchmarks | | | Video General Benchmarks | | | Avg. |
|---|---|---|---|---|---|---|---|---|
| | | VSI-Bench | VideoMMMU | MMVU | MVBench | TempCompass | VideoMME | |
| GPT-4o Hurst et al. (2024) | - | 34.0 | 61.2 | 75.4 | - | - | 71.9 | - |
| VideoLLaMA2-7B Cheng et al. (2024) | - | - | - | 44.8 | 54.6 | - | 47.9 | - |
| LongVA-7B Zhang et al. (2024b) | - | 29.2 | 23.9 | - | - | 56.9 | 52.6 | - |
| VILA-1.5-8B Lin et al. (2024) | - | 28.9 | 20.8 | - | - | 58.8 | - | - |
| LLaVA-OneVision-7B Li et al. (2024b) | - | 32.4 | 33.8 | 49.2 | 56.7 | - | 58.2 | - |
| Baseline (Qwen2.5-VL-7B) | 16 | 27.7 | 47.8 | 59.2 | 57.4 | 72.2 | 53.1 | 52.9 |
| Video-R1-SFT Feng et al. (2025) | 16 | 31.8 | 47.4 | 61.3 | 59.4 | 69.2 | 52.8 | 53.7 |
| Video-ToC-SFT (Ours) | 16 | 34.8 | 46.5 | 65.3 | 63.3 | 72.8 | 56.6 | 56.6 |
| TinyLLaVA-Video-R1 Zhang et al. (2025b) | 16 | - | - | 46.9 | 49.5 | - | 46.6 | - |
| VideoChat-R1 Li et al. (2025a) | 16 | 28.9 | 48.7 | 65.8 | 64.2 | 73.5 | 57.7 | 56.5 |
| Video-R1 Feng et al. (2025) | 16 | 34.6 | 49.8 | 64.2 | 62.7 | 72.6 | 57.4 | 56.9 |
| **Video-ToC** (Ours) | 16 | **35.3** | **50.5** | **66.1** | **65.0** | **73.8** | **58.6** | **58.2** |
| Baseline (Qwen2.5-VL-7B) | 32 | 30.1 | 48.1 | 60.0 | 59.0 | 72.6 | 56.6 | 54.4 |
| Video-R1-SFT Feng et al. (2025) | 32 | 33.3 | 49.4 | 63.5 | 60.5 | 69.9 | 55.4 | 55.3 |
| Video-ToC-SFT (Ours) | 32 | 35.8 | 47.1 | 65.4 | 65.3 | 73.7 | 59.8 | 57.9 |
| Video-R1 Feng et al. (2025) | 32 | 35.8 | **52.3** | 63.8 | 63.9 | 73.2 | 59.3 | 58.1 |
| **Video-ToC** (Ours) | 32 | **36.4** | 51.3 | **66.1** | **66.3** | **74.2** | **61.2** | **59.3** |

Table 2: Accuracy comparison on VideoHallucer Wang et al. (2024c) benchmark. "Avg." denotes average accuracy of the five task categories.

| Method | Object-Relation | Temporal | Semantic Detail | Factual | Non-factual | Avg. |
|---|---|---|---|---|---|---|
| Baseline (Qwen2.5-VL-7B) | 61.5 | 44.0 | 67.5 | **25.5** | 54.0 | 50.5 |
| Video-R1 | 54.5 | 39.5 | 62.0 | 23.5 | 47.5 | 45.4 |
| **Video-ToC** (Ours) | **66.0** | **45.0** | **74.0** | 20.0 | **54.5** | **51.9** |

Table 3: Performance comparison of different training strategies.

| Method | MMVU | MVBench | VideoMME |
|---|---|---|---|
| Baseline (Qwen2.5-VL-7B) | 59.2 | 57.4 | 53.1 |
| Baseline + GRPO | 63.8 | 61.1 | 54.3 |
| Baseline + SFT | 65.3 | 63.3 | 56.6 |
| Baseline + SFT + GRPO | **66.1** | **65.0** | **58.6** |

on both the reasoning and general benchmarks, which reveals the efficacy and generalizability of our constructed datasets and training strategies.

Table 2 evaluates the hallucination risks across different models. Compared to the baseline, Video-R1 exhibits more severe hallucination on most task categories. This validates that the rationales annotated by Video-R1, which are freely generated by a much more powerful model, are not suitable for the base model to learn and imitate. The reason is that these rationales are not appropriately tailored to the perceptual ability of the base model. As a consequence, the model tends to answer questions primarily by relying on its language knowledge rather than extracting key visual cues from the videos, thereby leading to more severe hallucination.

### 4.3 ABLATION STUDY

**The necessity of SFT cold start.** To investigate the effect of SFT cold start using the proposed Video-ToC-SFT-1k dataset, we skip the cold start stage and directly apply GRPO training using the proposed reasoning-demand reward to the baseline model, on the Video-ToC-RL-2k dataset. As shown in Table 3, the performance gains of 'Baseline + GRPO' are relatively small across all benchmarks, which may stem from the model's limited reasoning capacity for video understanding tasks. In contrast, the SFT cold start utilizing our Video-ToC-SFT-1k dataset equips the model with a reasoning paradigm that progressively identifies critical visual cues for better analyzing the question, which is more effective than the self-explored reasoning strategies. Consequently, the model after SFT cold start (termed as 'Baseline + SFT') significantly outperforms the variant trained exclusively via GRPO. Additionally, the reasoning strategies introduced by our constructed Video-ToC rationales can be further enhanced through subsequent GRPO training with the proposed reasoning-demand reward, leading to extra performance improvements.

Table 4: Effect of tree-guided visual cue localization. "Single-Cue-SFT" and "Tree-of-Cue-SFT" denote SFT using the Video-SingleCue-SFT-1k dataset and Video-ToC-SFT-1k dataset, respectively.

| Method | MMVU | MVBench | VideoMME |
|---|---|---|---|
| Baseline | 59.2 | 57.4 | 53.1 |
| Baseline + Single-Cue-SFT | 61.8 | 61.0 | 54.5 |
| Baseline + Tree-of-Cue-SFT | **65.3** | **63.3** | **56.6** |

Table 5: Effect of reasoning-demand reward.

| Method | MMVU | MVBench | VideoMME |
|---|---|---|---|
| Video-ToC-SFT | 65.3 | 63.3 | 56.6 |
| Video-ToC-SFT + GRPO w/ Vanilla Reward | 65.6 | 64.2 | 57.6 |
| Video-ToC-SFT + GRPO w/ Reasoning-demand Reward | **66.1** | **65.0** | **58.6** |

**Effect of tree-guided visual cue localization.** A key design of our method is introducing the tree structure to help annotate the Video-ToC rationales with the reasoning pattern of tree-guided visual cue localization, ultimately obtaining the Video-ToC-SFT-1k dataset. To validate its effectiveness, we construct an analogous SFT dataset where there is only a single step of cue localization in the rationales, and name it as Video-SingleCue-SFT-1k. Specifically, we use only the descriptions of the selected key clips to prompt the LLM to generate the rationales, thereby eliminating the need for constructing a tree. The sole difference between this dataset and our Video-ToC-SFT-1k lies in their rationale pattern, where the rationales in Video-SingleCue-SFT-1k follow a style of directly locating the cue and then analyzing the question. As shown in Table 4, using our Video-ToC-SFT-1k dataset for SFT achieves superior performance on all benchmarks, demonstrating the advantage of introducing tree for annotating rationales with a tree-guided visual cue localization pattern.

**Effect of reasoning-demand reward.** Table 5 presents an ablation study of the reward-design choices during GRPO training. The formal descriptions of vanilla reward and our reasoning-demand reward are demonstrated in Equation equation 2 and Equation equation 3, respectively. Note that for GRPO training with vanilla accuracy reward, the reasoning demand used in Equation equation 5 is set as a constant value of 1 (i.e., $\gamma = 1$). As shown in Table 5, the proposed reasoning demand-driven reward mechanism consistently improves accuracy on all benchmarks compared to conventional GRPO training which uses vanilla accuracy reward, highlighting the benefits of tailoring incentive levels to questions with varying reasoning demands.

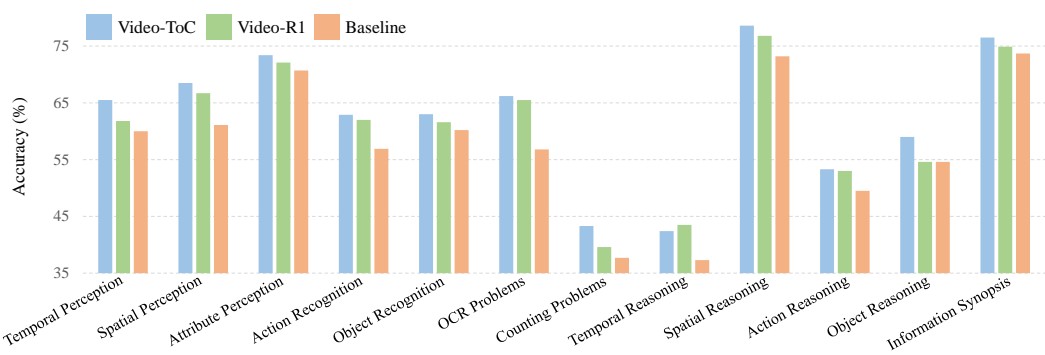

Figure 3: Quantitative analysis of task improvements on VideoMME benchmark.

## 4.4 VISUALIZATION ANALYSIS

**Quantitative results.** To assess the effect of Video-ToC on the improvements of specific tasks, we conduct a statistical analysis of task category results on the VideoMME benchmark, comparing against the baseline and Video-R1, as shown in Figure 3. Notably, Video-Toc demonstrates significant improvements over the baseline across all tasks. It also outperforms Video-R1 on most categories, particularly the 'Temporal Perception', 'Counting Problems', and 'Object Reasoning' tasks, which demonstrates that our method can effectively enhance both the perception and reasoning capabilities of the model.

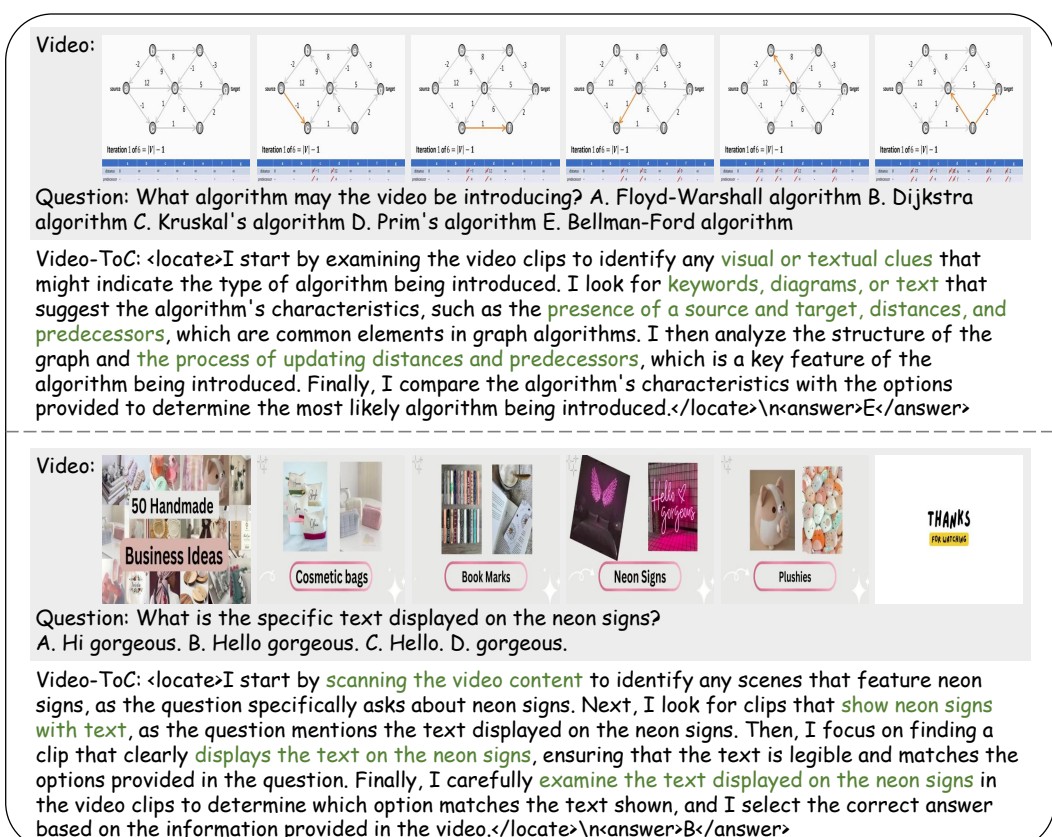

Figure 4: Two examples of Video-ToC's output from MMVU (top) and VideoMME (bottom).

**Qualitative results.** We present two examples of our Video-ToC's output, respectively drawn from the video reasoning benchmark MMVU and the video general benchmark VideoMME, as shown in Figure 4. For both questions, Video-ToC employs a step-by-step approach to locate key visual cues for reasoning. Specifically, when addressing the reasoning-based question, Video-ToC first deduces critical visual cues that help solving the question based on its knowledge and searches for them progressively. In contrast, for the perception-based question, it meticulously scans and examines key visual cues according to the queries in the question, reasoning primarily based on the semantic information in the video rather than its knowledge. These examples demonstrate the effectiveness and flexibility of Video-ToC's reasoning strategies across various question types.

## 5 CONCLUDING REMARKS

**Summary.** We propose Video-ToC, a novel video reasoning framework that incorporates a tree-guided visual cue localization mechanism and a reasoning-demand-based reward strategy. To endow the model with robust reasoning capabilities, we develop an automatic data annotation pipeline to construct two high-quality datasets: Video-ToC-SFT-1k and Video-ToC-RL-2k, dedicated to supervised fine-tuning and reinforcement learning, respectively. Extensive experiments across six video understanding benchmarks and one video hallucination benchmark validate the efficacy of our approach, demonstrating consistent performance improvements and hallucination mitigation.

**Limitations and future work.** Our current experiments employ uniform sampling with 16 or 32 input frames. Future work will involve exploring the effect of increasing input frames and employing different frame sampling strategies. Additionally, while we have focused on curating video training data, numerous high-quality image reasoning datasets remain underexplored. We aim to devise methodologies for leveraging image-video hybrid reasoning data to enhance the model's video reasoning capabilities.

## 6 ETHICS STATEMENT

This work adheres to the ICLR Code of Ethics. Our study does not involve human subjects, personal data, or sensitive attributes, and thus does not raise direct privacy or security concerns. The datasets used are publicly available, and we strictly follow their licenses and usage guidelines. No proprietary or non-consensually collected data were employed. We declare no conflicts of interest and ensure compliance with ethical standards in data handling and research integrity.

## 7 REPRODUCIBILITY STATEMENT

We have made extensive efforts to ensure the reproducibility of Video-ToC. Sec. 3 and Sec. 4 in the main paper present the dataset construction pipeline, model choice, training objectives, and evaluation protocols. Detailed hyperparameter settings, implementation details, and prompt designs are included in Sec. 4.1 of the main paper and in Secs. E-G of the supplementary material. We will release all code, models, and datasets following the completion of the blind review process.

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

# Video-ToC: Video Tree-of-Cue Reasoning

## Supplementary Material

## A  AN EXAMPLE OF VIDEO-TOC RATIONALE ANNOTATION PIPELINE

In Figure 5, we present an example of the Video-ToC rationale annotation pipeline.

We first build a Segment Tree based on the segmented video clips. Note that the core design of Video-ToC lies in the hierarchical reasoning strategy enabled by tree-guided visual cue localization, rather than the specific tree structure like a complete binary tree, which is just a practical choice for systematic video decomposition and trajectory generation. Any tree structure that allows for a multi-level decomposition of video content (enabling coarse-to-fine localization of visual cues) would align with the goals of Video-ToC. The Segment Tree is merely a straightforward instantiation of this idea.

Then, an LLM (Llama-3.3-70B-Instruct Grattafiori et al. (2024)) selects the relevant video clips using their captions generated by an MLLM (Qwen2.5-VL-7B Bai et al. (2025)). After that, the reasoning trajectory is derived by performing backtracking from the leaf nodes (selected video clips) to the root node (the whole video). Note that when multiple clips are found to be relevant, the reasoning trajectory forms a subtree rather than a single chain or path (see Figure 5 for an example).

Next, we extract video segments from each layer of this trajectory (i.e., the red nodes in Figure 5) and concatenate them to form the 'Video Compilations'. These compilations are deduplicated as 'Visual Cues' and then captioned by an MLLM (Qwen2.5-VL-7B Bai et al. (2025)). Finally, an LLM (Llama-3.3-70B-Instruct Grattafiori et al. (2024)) integrates these 'Visual Cue Descriptions' with the corresponding question-answer pair to generate the Video-ToC rationale. Because concatenation linearizes the trajectory into a chain, the LLM no longer needs to process the original tree structure and can instead interpret the visual cues as a single reasoning path during summarization.

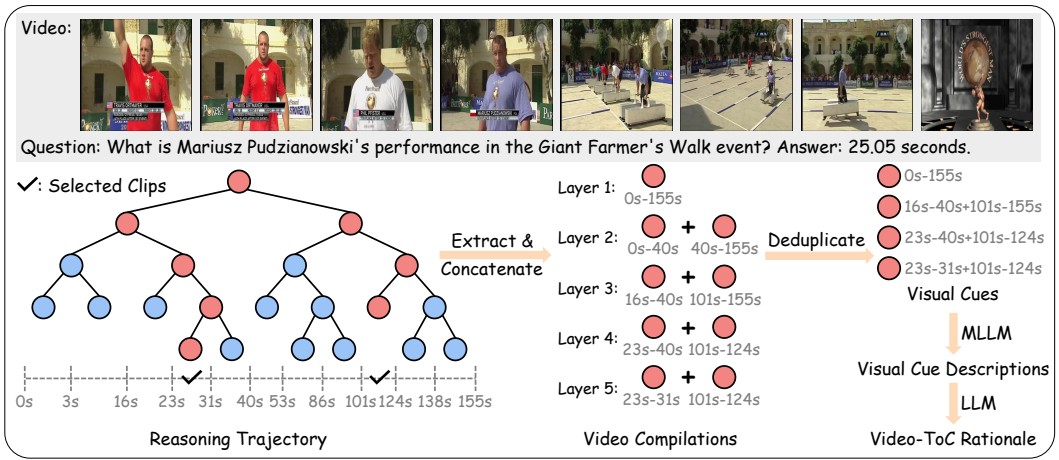

Figure 5: Additional illustrations of Video-ToC rationale annotation pipeline.

## B  PERFORMANCE COMPARISON UNDER MORE INPUT FRAMES

We increase the number of input frames to 64 during evaluation, and compare with the baseline and Video-R1 in Table 6. Consistent with the results under 16- and 32-frame settings shown in Table 1 in the paper, our Video-ToC-SFT still outperforms both the baseline and the SFT model of Video-R1 by a large margin on most benchmarks. Furthermore, after RL training, our final model Video-ToC achieves the best overall performance and significantly surpasses Video-R1, demonstrating the effectiveness and generalizability of our method.

Table 6: Accuracy comparison using 64 frames as input. "Avg." denotes average accuracy of the six benchmarks.

| Method | Frames | Video Reasoning Benchmarks | | | Video General Benchmarks | | | Avg. |
| --- | --- | --- | --- | --- | --- | --- | --- | --- |
| | | VSI-Bench | VideoMMMU | MMVU | MVBench | TempCompass | VideoMME | |
| Baseline | 64 | 31.4 | 50.4 | 60.0 | 59.2 | 72.9 | 59.6 | 55.6 |
| Video-R1-SFT | 64 | 34.8 | 49.4 | 61.6 | 60.6 | 70.0 | 58.8 | 55.9 |
| Video-ToC-SFT (Ours) | 64 | 37.6 | 48.3 | 65.4 | 65.7 | 73.9 | 61.1 | 58.7 |
| Video-R1 | 64 | 37.1 | **52.4** | 63.8 | 64.8 | 73.2 | 61.4 | 58.8 |
| **Video-ToC** (Ours) | 64 | **38.6** | 51.0 | **66.5** | **66.4** | **74.2** | **62.6** | **59.9** |

Table 7: Effect of format reward. "Correct Format" denotes the percentage of responses that adhere to the specified format.

| Method | MMVU | MVBench | VideoMME | Correct Format |
| --- | --- | --- | --- | --- |
| GRPO w/o format reward | 66.1 | 65.0 | 58.6 | 100.0 |
| GRPO w/ format reward | 66.2 | 64.7 | 58.6 | 100.0 |

## C  ADDITIONAL DISCUSSIONS ON FORMAT REWARD

The format reward is typically used to guide the model to put its thinking process between the "<think>" and "</think>" tags and place its answer between the "<answer>" and "</answer>" tags. However, during GRPO training, we only apply the proposed reasoning-demand reward without requiring the format reward. This is because we incorporate detailed formatting guidelines within the prompts (see Figure 6), and the model after the SFT stage (i.e., Video-ToC-SFT) can adhere to the specified format well enough. Moreover, if a response fails to follow this format (e.g., the answer is not placed within the "<answer>" and "</answer>" tags), the reasoning-demand reward for this response may be zero even if the answer is correct, which implicitly enforces the model to follow the specified format. As shown in Table 7, applying format reward reveals negligible effect on the performance, and the model successfully follows the specified format for all test samples. Therefore, we remove the unnecessary format reward in GRPO training.

## D  ADDITIONAL DISCUSSIONS ON ADVANTAGE CALCULATION

In GRPO Shao et al. (2024b), the advantages of different responses within a group are calculated by normalizing their rewards (see Equation (4) in the paper). Since the reward for each response consists solely of the reasoning-demand reward which is a binary function, adjusting the values of the reasoning-demand rewards for different questions will have no effect. Specifically, for a question with a reasoning demand of $\gamma$, the reward of each response within the group can only be either $\gamma$ or 0. Suppose a group contains $G$ responses, where $x$ ($0 \leq x \leq G$) of them correctly answer the question. The advantages of these responses are as follows:

$$A_{\text{correct}} = \frac{r_{\text{correct}} - \text{mean}(\{r_i\}_{i=1}^{G})}{\text{std}(\{r_i\}_{i=1}^{G})} \tag{7}$$

$$= \frac{\gamma - \frac{x\gamma}{G}}{\sqrt{\frac{1}{G-1}\left(x \cdot (\gamma - \frac{x\gamma}{G})^2 + (G-x) \cdot (0 - \frac{x\gamma}{G})^2\right)}} \tag{8}$$

$$= \sqrt{\frac{(G-1) \cdot (G-x)}{Gx}} \quad (0 < x \leq G), \tag{9}$$

where $r_{\mathrm{correct}}$ and $A_{\mathrm{correct}}$ respectively denote the reward and advantage of the correct response. Similarly, the responses with wrong answers will obtain the advantages of:

$$A_{\mathrm{wrong}} = \frac{r_{\mathrm{wrong}} - \mathrm{mean}(\{r_i\}_{i=1}^{G})}{\mathrm{std}(\{r_i\}_{i=1}^{G})} \tag{10}$$

$$= \frac{0 - \frac{x\gamma}{G}}{\sqrt{\frac{1}{G-1}\left(x \cdot (\gamma - \frac{x\gamma}{G})^2 + (G-x) \cdot (0 - \frac{x\gamma}{G})^2\right)}} \tag{11}$$

$$= -\sqrt{\frac{x \cdot (G-1)}{G \cdot (G-x)}} \ (0 \leq x < G), \tag{12}$$

where $r_{\mathrm{wrong}}$ and $A_{\mathrm{wrong}}$ represent the reward and advantage of the wrong response, respectively. We can observe that the advantages used for optimizing the model are irrelevant to the specific value of $\gamma$, which we adjust for different questions. Therefore, we multiply the original advantages by the reasoning demand $\gamma$ (see Equation (5) in the paper) to tailor the magnitudes of advantages for questions with different reasoning demands.

## E  ADDITIONAL DETAILS OF RL DATA CONSTRUCTION

We randomly select a subset of the LLaVA-Video-178K dataset Zhang et al. (2024c) and annotate the reasoning demands to construct our RL training dataset. The LLaVA-Video-178K dataset includes both open-ended and multiple-choice QA (question and answer) items. However, we choose the multiple-choice QA items exclusively, as they tend to yield more accurate reward signals for RL. For each video QA, we employ Qwen2.5-VL-7B Bai et al. (2025) to directly answer the question across 8 independent trials ($M = 8$) and record the count of correct predictions as $\alpha$. Then, we calculate the difficulty score and reasoning demand for this question as $1 - \frac{\alpha}{M}$ and $e^{-\frac{\alpha}{M}}$, respectively. To avoid the computed advantages from being all zeros, we exclude questions that are too easy or too hard to answer, i.e., questions with difficulty scores below 0.2 or above 0.8 are discarded.

## F  PROMPT DETAILS

### F.1  PROMPT FOR TRAINING AND INFERENCE

The prompt for both model training and inference is provided in Figure 6. Following Video-R1 Feng et al. (2025), the last sentence of the prompt serves as the 'task instruction' to guide the model in adhering to the formats of different types of questions.

### F.2  PROMPT FOR KEY CLIPS SELECTION

During Step 1 of SFT data construction, we employ Llama-3.3-70B-Instruct Grattafiori et al. (2024) to select the key clips that are essential for answering the provided question. The prompt is detailed in Figure 7.

### F.3  PROMPT FOR LOW-QUALITY CUES FILTERING

In Step 3 of SFT data construction, we employ Llama-3.3-70B-Instruct Grattafiori et al. (2024) to filter out samples where the visual cues from the final step are insufficient to derive the answer to the question, using the prompt in Figure 8.

### F.4  PROMPT FOR VIDEO-TOC RATIONALE GENERATION

The Video-ToC rationale is generated by prompting Llama-3.3-70B-Instruct Grattafiori et al. (2024) to summarize the processed reasoning trajectory. The specific prompt is provided in Figure 9. To obtain rationales with a step-by-step style, we instruct the LLM to follow the specified format: "Step 1: ... Step 2: ... Step 3: ...". Subsequently, we remove the rigid "Step k:" structure to make the rationales more natural.

## G   ADDITIONAL IMPLEMENTATION DETAILS

For both the SFT and RL stages, we employ the Adam optimizer with a learning rate set to 5e-7 to train the model for 1 epoch. The SFT is conducted using the LlamaFactory codebase Zheng et al. (2024), while the RL is performed using the EasyR1 codebase Zheng et al. (2025). Specifically, the SFT stage runs for 125 steps with a batch size of 8, whereas the RL stage runs for 500 steps with a batch size of 4.

## H   EXAMPLES OF THE VIDEO-TOC-SFT-1K DATASET

We provide two qualitative examples of the Video-ToC rationales in our Video-ToC-SFT-1k dataset, as shown in Figure 10. Both rationales in the annotated answers demonstrate the process of locating video clips that are increasingly helpful for solving the question and reaching the answer.

## I   BROADER IMPACT

This paper presents work whose goal is to advance the field of machine learning. Although our work may carry a range of societal implications, we do not identify any that require special emphasis here.

## J   USE OF LARGE LANGUAGE MODELS

We used large language models (LLMs) solely to aid in polishing the writing of this paper. The research ideas, methodology, experiments, analyses, and conclusions were entirely developed and carried out by the authors. LLMs did not contribute to research ideation, experiment design, or result interpretation. All authors take full responsibility for the content of the paper.

---

**Prompt for Training and Inference**

---

{Question}
First, progressively locate video clips that are increasingly helpful for answering the question, and then provide your final answer. Put your detailed locating process between the <locate> </locate> tags, and your final answer between the <answer> </answer> tags. {Task Instruction}

Task Instruction:
"multiple choice": "Provide only the single option letter (e.g., A, B, C, D, etc.) within the <answer> </answer> tags."
"numerical/regression": "Provide the numerical value (e.g., 42 or 3.14) within the <answer> </answer> tags."

---

Figure 6: Prompt for training and inference.

---

**Prompt for Key Clips Selection**

---

### Task:
You are an excellent problem solver with a strong ability to comprehend and analyze long-form video content. There is a long video that has been split into multiple semantically coherent clips to help you understand. You are provided with the detailed description for each clip, and a question-answer pair based on this long video. Please carefully understand this long video based on the detailed descriptions for all clips, along with the question-answer pair. And reason how to solve this question using the information provided in the video to arrive at the correct answer. Based on your reasoning process, identify which clips are essential for answering the question.

### Guidelines:
The information provided to you regarding the long video is given in JSON format, which includes the count of clips, the index and detailed description for each clip, and a question-answer pair based on this long video. You should only provide the indices of your selected clips. No need to explain.

### Output Format:
It is critical that you respond only with the exact, parseable JSON and not any preamble, explanation, or anything else outside of the valid JSON as your outputs will be fed directly to a JSON parser to go into a downstream application. Do not include any markup like ```json or anything else that would break our ability to parse the response. This is critical, after you are done reasoning and before you respond, ensure that your response is exactly JSON parseable. You must respond with a JSON array that matches the following schema:
[<index_1>, <index_2>, ..., <index_N>]

Please provide the indices of the essential clips for the following video clip descriptions and corresponding question-answer pair:
{Video Clip Descriptions}; {Question}; {Answer}

---

Figure 7: Prompt for key clips selection.

---

**Prompt for Low-quality Cues Filtering**

---

I will provide you with a question-answer pair, along with a detailed description of a video. You need to judge whether the video content is sufficient to lead to the answer to the question. If so, respond with "Yes"; otherwise, respond with "No". No need to explain. Please provide your judgement for the following question-answer pair and video content:
{Question}; {Answer}; {Cues}

---

Figure 8: Prompt for low-quality cues filtering.

---

**Prompt for Video-ToC Rationale Generation**

You are an excellent video assistant with a strong ability to comprehend and analyze long-form video content, and you are watching a long video. I will provide you with a question-answer pair and explain the process of locating video clips that are increasingly helpful for solving the question and reaching the answer. Please summarize the locating process in the first-person tone, demonstrating the step-by-step method of how to locate the most important clip for the given question. While you are summarizing, act as if you can only see the entire video and question, and you are unaware of the provided video clip descriptions and the given answer. Your response should be concise, presented in a single paragraph, and follow this format: "Step 1: ... Step 2: ... Step 3: ...". Note that the number of steps in your response MUST equal the number of steps in the provided locating process. Please provide your summarized locating process for the following data: {Question}; {Answer}; {Reasoning Trajectory}

---

Figure 9: Prompt for Video-ToC rationale generation.

Video: 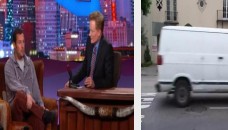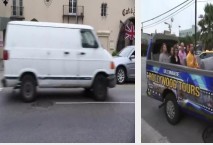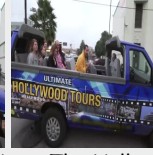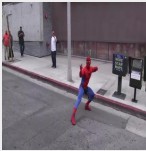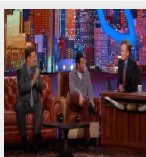

Question: Which landmark does the tour van pass by?\nA. The Hollywood Sign\nB. The Griffith Observatory\nC. A Spider-Man performer\nD. The Walk of Fame\nFirst, progressively locate video clips that are increasingly helpful for answering the question, and then provide your final answer. Put your detailed locating process between the <locate> </locate> tags, and your final answer between the <answer> </answer> tags. Provide only the single option letter (e.g., A, B, C, D, etc.) within the <answer> </answer> tags.

Answer: <locate>I start by examining the video content to identify any scenes that involve a tour or a van passing by notable landmarks, as this is the central theme of the question. Next, I look for clips that feature a tour bus or van with a clear label or signage, such as \"Ultimate Hollywood Tours,\" to establish a connection to the tour mentioned in the question. Then, I analyze the clips to find scenes where the tour bus passes by or is near recognizable landmarks or characters, which could be the answer to the question. Finally, I focus on identifying a specific landmark or character that the tour van passes by, looking for clear visuals or context clues that match one of the provided options, to determine the most accurate answer to the question.</locate>\n<answer>C</answer>

- - - - - - - - - - - - - - - - - - - - - - - - - - - - - - - - - - - - - - - - - - - - -

Video: 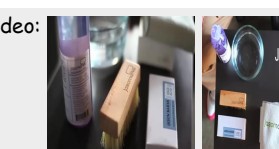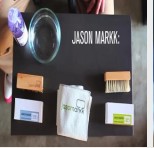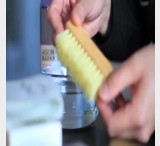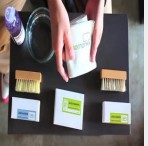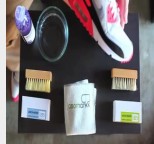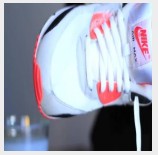

Question: Which brand of sneakers is being cleaned in the video?\nA. Adidas\nB. Puma\nC. Reebok\nD. Nike Air Max\nFirst, progressively locate video clips that are increasingly helpful for answering the question, and then provide your final answer. Put your detailed locating process between the <locate> </locate> tags, and your final answer between the <answer> </answer> tags. Provide only the single option letter (e.g., A, B, C, D, etc.) within the <answer> </answer> tags.

Answer: <locate>I start by examining the video clips to identify any that show a shoe cleaning process, looking for details such as the type of shoes being cleaned and the cleaning products used. Next, I narrow down my search to clips that not only show the cleaning process but also provide a clear view of the sneakers being cleaned, including any visible logos or branding that could help identify the brand. Finally, I focus on finding a clip that explicitly shows the brand of the sneakers, such as a close-up of the shoe's label or a clear shot of the brand's logo, to determine the correct answer to the question about the brand of sneakers being cleaned.</locate>\n<answer>D</answer>

Figure 10: Two qualitative examples of the Video-ToC-SFT-1k dataset.