# OpenReview forum: "Video-ToC: Video Tree-of-Cue Reasoning"
_ICLR.cc/2026/Conference — ICLR 2026 Conference Withdrawn Submission_

### Official Review · Reviewer_49M5 · 2025-10-20

**Soundness:** 3
**Presentation:** 2
**Contribution:** 3
**Rating:** 6
**Confidence:** 5

**Summary:**

The paper proposes Video-ToC, a video reasoning framework designed to enhance the performance and interpretability of Video Large Language Models (Video LLMs) by introducing a tree-guided visual cue localization mechanism and a dynamic, reasoning-demand-based reward for reinforcement learning. The approach includes a novel rationale annotation pipeline to generate progressive tree-structured reasoning samples (Video-ToC-SFT-1k) and an RL training set with automatically estimated reasoning demand (Video-ToC-RL-2k). The model is evaluated on six video understanding and one hallucination benchmark, showing improvements over previous state-of-the-art methods in both overall performance and hallucination mitigation.

**Strengths:**

**1. Innovation of Tree of Cues:**
The paper introduces a Tree of Cues framework that models reasoning as a progressive traversal from coarse to fine video segments, offering a structured, interpretable process for visual cue localization. This tree-based formulation, clearly depicted in the pipeline diagram (Figure 2) and illustrated through supplementary examples (Figure 5), enhances both interpretability and the alignment between visual perception and reasoning steps.

**2. Comprehensive Experimental Evaluation**:
The paper presents a comprehensive experimental evaluation, covering a broad set of benchmarks (Tables 1–6) and showing consistent, non-trivial performance improvements over strong baselines in both video reasoning and hallucination mitigation tasks. Ablation studies (Table 4 for tree vs. single-cue structures; Table 5 for reward design) and detailed visual analyses (Figures 3, 4, 5, 10) further clarify where and how the proposed components contribute to these gains.

**Weaknesses:**

**1. Marginal Reward Novelty:**
The proposed Reasoning-Demand Reward adaptively adjusts incentives according to the estimated complexity of each query. However, it functions as a heuristic reweighting of RL rewards derived from an external MLLM, without theoretical grounding or sufficient empirical justification. The design closely resembles prior difficulty-aware or confidence-weighted RL schemes.

**2. Tiny Scale of Dataset:**
The “Video-ToC-SFT-1k” and “Video-ToC-RL-2k” are extremely small (3k total training samples), such a limited scale is insufficient to convincingly demonstrate generalizable reasoning improvements in video LLMs. The observed performance boost may stem primarily from strong prompt supervision or curated data quality rather than genuine advancement in reasoning capability. Scaling and data-size control experiments are required to validate whether the framework truly improves reasoning or merely benefits from targeted supervision.

**3. “Tree” Advantage Remains Ambiguous:**
The Single-Cue vs. Tree-of-Cue ablation (Table 4) shows gains, but it still entangles captioner quality, LLM summarization, and clip selection. There is no random clips or manually verified clips (if possible) to compare with, nor an ablation of backtracking depth.

**4. Missing Ablation of Advantage Calculation:**
The paper introduces a modified advantage computation that multiplies normalized group advantages by the reasoning-demand coefficient (Eq. 5), but there is no ablation or sensitivity analysis isolating this design choice.

**5. Trade-offs between Gains and Compute Costs:**
RL with reasoning-demand rewards and tree-structured preprocessing seems computationally expensive.

**Questions:**

1. What happens if you scale the training data beyond 3k samples — for example, to 5k, 10k, or 20k examples? Does the improvement persist or saturate?
2. Could you provide a more detailed qualitative analysis of failure cases?
Specifically, which categories of reasoning remain problematic — e.g., temporal reasoning, causal inference, spatial relations, or fine-grained attribute localization? Understanding what fails could clarify where the tree-guided cue localization or reasoning-demand reward are still insufficient.
3. Could you show how sensitive the model performance is to multiply GRPO advantage  with reasoning-demand coefficient?
4. Could add clip-selection ablations if possible, such as (i) random clips; (ii) top-K by caption similarity to the question; (iii) oracle (human/GT) when available. If varing tree depth and branching influence the model performance?

---

### Official Review · Reviewer_HHkv · 2025-10-27

**Soundness:** 3
**Presentation:** 3
**Contribution:** 3
**Rating:** 4
**Confidence:** 3

**Summary:**

This paper proposes Video-ToC, a two-stage framework to improve the reasoning capabilities and reduce hallucinations in Video Large Language Models. The authors argue that existing models often rely on pretrained language priors rather than grounding their reasoning in the video's visual content, especially when weaker models try to imitate rationales from stronger ones. The proposed solution consists of: 1) A sft stage using a novel tree-guided visual cue localization mechanism. This involves a complex automated pipeline to generate a new dataset, Video-ToC-SFT-1k, where rationales guide the model to find key visual cues in a step-by-step manner. 2) A rl stage using a reasoning-demand reward, which dynamically adjusts the reward value based on an estimation of the question's reasoning complexity. This is used to train on another new dataset Video-ToC-RL-2k. The authors demonstrate that their final model, Video-ToC, achieves modest performance gains over prior state-of-the-art methods on several video understanding and hallucination benchmarks.

**Strengths:**

1. The paper correctly identifies an important and challenging problem in video reasoning: the mismatch between a model's perceptual capabilities and the free-form rationales they are often trained on, which can lead to increased hallucination.

2. The core idea of explicitly training the model to first locate visual evidence (using the <locate> tags) before answering is intuitive and directly addresses the identified problem.

3. The proposed framework does show some success in mitigating certain types of hallucinations when compared to the Video-R1 method (Table 2).

**Weaknesses:**

1. The primary weakness of this paper is that the final performance improvement over the SOTA baseline (Video-R1) is marginal. For example, on the 16-frame setting, the average accuracy improvement is 1.3 points (58.2 vs 56.9). On the 32-frame setting, the gain is 1.2 points (59.3 vs 58.1). While consistent, these gains are incremental and do not seem to justify the significant complexity of the proposed solution.

2. The paper's main contribution is not a novel model architecture but rather a highly complex, multi-stage data generation pipeline. This pipeline requires a video splitter, an MLLM for captioning, an LLM for key-clip selection (based on the captions), and another LLM for rationale summarization . This approach feels more like a "brute-force" data engineering effort than an elegant or generalizable modeling innovation. This high complexity also raises serious concerns about reproducibility and the practical cost of applying this method.

3. The paper's claim of mitigating hallucination is not fully supported by the data. While the average score in Table 2 improves slightly over the baseline, the model performs significantly worse in the "Factual" hallucination category. Video-ToC scores 20.0, which is lower than both the Baseline (25.5) and Video-R1 (23.5). This is a critical flaw, as it suggests the model's new reasoning style may be less grounded in factual, extrinsic knowledge than the very methods it aims to improve upon.

4. The "reasoning-demand reward" is presented as a key innovation. However, the ablation study in Table 5 reveals that its contribution to the final performance is also incremental (e.g., a 0.5 point gain on MMVU from 65.6 to 66.1, and a 0.8 point gain on MVBench from 64.2 to 65.0). This appears to be a minor modification to the reward calculation within the existing GRPO framework , not a fundamental advance. In fact, the ablation in Table 3 shows that the SFT stage alone ("Baseline + SFT") provides the vast majority of the performance gain, suggesting the complex RL stage adds very little.

**Questions:**

1. Could the authors explain the poor performance on the "Factual" hallucination task (Table 2)? Why does a method designed to improve visual grounding and reduce hallucination perform significantly worse on factual questions than both the baseline and Video-R1?

2. The SFT pipeline's reliance on an LLM selecting key clips based only on MLLM-generated text captions seems fragile. What is the pipeline's failure rate if the MLLM fails to caption a subtle but critical visual-only event (e.g., a small object being dropped)?

3. Given that the SFT stage ("Baseline + SFT") achieves 65.3 on MMVU and the final model ("Baseline + SFT + GRPO") achieves 66.121, the complex RL stage (including the novel reward) contributes only a 0.8-point gain. Does this small improvement justify the added complexity, data annotation (for $R_{rd}$), and training costs of the entire RL stage?

---

### Official Review · Reviewer_AwZb · 2025-10-28

**Soundness:** 2
**Presentation:** 3
**Contribution:** 3
**Rating:** 4
**Confidence:** 3

**Summary:**

This paper trains video models to look first, then answer. Instead of guessing from language priors, it has the model walk through a coarse-to-fine “find the evidence” routine, then give the final reply. The authors build a small, automated dataset that shows this step-by-step behavior, and then add a light RL phase that pushes the model to use that behavior on harder questions. In tests, this setup beats strong baselines and cuts down on hallucinations, suggesting that making the model explicitly hunt for visual clues helps.

**Strengths:**

1. Introduce the tree-of-thoughts idea into the video domain.

2. Introduce an automated pipeline that injects a coarse-to-fine “look-before-answer” tree-of-cue with only LLM supervision.

**Weaknesses:**

1. The paper doesn’t provide a theoretical framework explaining why hierarchical ToC supervision should improve credit assignment or reduce hallucination

2. Generating \<locate\> at inference increases tokens and wall-clock time; throughput and real-time viability aren’t fully characterized.

**Questions:**

1. In the “multi-step vs. single-step” comparison, is there an equal-length control to rule out gains purely from longer \<locate\> rationales (i.e., more tokens) rather than the hierarchical structure itself?

2. Is the \<locate\> block’s verifiability quantified—for example, by time-stamped, reproducible references to the supporting video segments?

3. In GRPO, does estimating problem difficulty (Reasoning Demand) via direct-answer accuracy introduce a bias relative to the accuracy achieved after ToC-style reasoning? Could $\gamma$ be estimated on-policy instead—i.e., directly from the $K$ group samples generated during GRPO for each prompt?

---

### Official Review · Reviewer_cpcX · 2025-11-10

**Soundness:** 4
**Presentation:** 3
**Contribution:** 3
**Rating:** 4
**Confidence:** 4

**Summary:**

This paper introduces Video-ToC, a new framework designed to enhance the reasoning capabilities of Video Large Language Models (VideoLLMs) and mitigate hallucinations. The authors identify that existing Video LLMs often struggle with complex reasoning tasks that require fine-grained visual perception, tending to rely on pre-trained knowledge rather than the video's content. To address this, Video-ToC proposes a "tree-of-cue" reasoning approach.
The core of their methodology consists of three main contributions:
1. A tree-guided visual cue localization mechanism: This method endows the model with a structured, hierarchical reasoning pattern to progressively narrow down on relevant visual information in a video.
2. A reasoning-demand reward mechanism: This dynamically adjusts the reward for reinforcement learning (RL) based on the complexity of the reasoning required for a given question, providing a more nuanced incentive for the model to develop effective reasoning strategies.
3. An automated annotation pipeline: This pipeline is used to construct two new datasets, Video-ToC-SFT-1k for supervised fine-tuning (SFT) and Video-ToC-RL-2k for RL training.

The authors demonstrate through extensive evaluations on six video understanding benchmarks and one video hallucination benchmark that Video-ToC surpasses baseline models and recent state-of-the-art methods.

**Strengths:**

**Novel and Intuitive Reasoning Framework**: The core idea of "tree-of-cue" reasoning, which mimics a hierarchical, coarse-to-fine human-like approach to analyzing complex scenes, is both novel and intuitive for video understanding. This structured approach to reasoning is a good solution for addressing how VideoLLMs struggle to analyze a video’s content. It is a departure from less constrained chain-of-thought methods and appears to be more grounded in the visual content.

**Innovative Reinforcement Learning Reward**: The proposed reasoning-demand reward is a clever way to address the varying difficulty of video reasoning tasks. By using the model’s uncertainty calibration as a metric for difficulty, the framework can more effectively incentivize the model to develop robust reasoning abilities rather than just relying on simple pattern matching.

**Strong Empirical Results**: The paper presents a comprehensive set of experiments on a wide range of benchmarks, including those focused on reasoning, general video understanding, and hallucination. The consistent outperformance of Video-ToC over baselines and prior work, as shown in Tables 1 and 2, provides strong evidence for the efficacy of the proposed method.

**Thorough Ablation Studies**: The paper includes a good set of ablation studies that effectively demonstrate the contributions of the key components of their framework, namely the SFT cold start, the tree-guided visual cue localization, and the reasoning-demand reward.

**Weaknesses:**

**Complexity and Scalability of Data Annotation**: The automated data generation pipeline, while innovative, is quite complex. This raises concerns about the computational cost and scalability of creating larger datasets. The process of building a segment tree, backtracking, and summarizing with LLMs may not be straightforward to replicate and could be a bottleneck for future work. More validation on this front would improve the reproducibility of this work. What do the extracted video segments look like, and are the curated ToC thoughts general or targeted towards solving the given questions, especially beyond qualitative observations?

**Limited Exploration of Tree Structure and Video Segmentation**: The paper primarily focuses on a binary tree structure for video segmentation based on shot boundary detection. It is not clear how well this approach would generalize to videos with very different temporal structures (e.g., long, continuous shots or rapidly changing scenes). The impact of different tree structures or more advanced video segmentation techniques on the final performance is not explored.

**Sensitivity of Reasoning-Demand Reward**: The reasoning-demand reward is calculated based on the performance of an MLLM over M=8 independent trials. The choice of M seems somewhat arbitrary, and the stability and reliability of this metric could be sensitive to both M and the specific MLLM used. A different MLLM might yield a different assessment of a question's reasoning demand.

**Limited Benefit from GRPO**: The different types of rewards are ablated in Table 5, compared to the baseline with no GRPO, where it seems GRPO based methods don’t provide much improvement over the baseline. This brings into question if RL is necessary here. For example, could the reasoning-demand be used for SFT as a weight in the NLL computation to prioritize harder examples instead? This ablation would be interesting, as it seems the SFT provides the bulk of the performance, which is not the case without SFT (as shown in Table 3).

As a general comment, the contribution of the “tree-of-cue” reasoning was difficult to grasp, as Tree-of-Cue reasoning implies the model is constructing such a hierarchy, which is not happening explicitly. If my understanding is correct, the reasoning traces are collected in such a way, but the reasoning is still a standard CoT with implicit RL shaping its thoughts to be hierarchical.

**Questions:**

**Regarding the Tree Structure**: What was the reasoning behind choosing a binary tree structure for video segmentation? Have you considered more flexible graph-based representations that might better capture the semantic relationships between non-adjacent video clips?

**On the Reasoning-Demand Reward**: How sensitive is the reasoning-demand reward to the choice of the model used for its estimation and the number of trials (M)? Would a more capable or less capable model for estimating the reasoning demand significantly alter the training dynamics and final performance?

**Generalizability of Learned Reasoning**: How well do the reasoning patterns learned from the Video-ToC-SFT-1k dataset generalize to video domains and question types not seen during training? Is there a risk that the model is overfitting to the specific "step-by-step localization" style of the generated rationales?

**Computational Overhead**: Could you provide more details on the computational cost of the data annotation pipeline? Specifically, how much time and resources are required to generate the Video-ToC-SFT-1k and Video-ToC-RL-2k datasets? I care mostly about how expensive it is to rerun this pipeline so many times for each hierarchy level (as far as I understand), such as the frame sampling rate (unless it's everything?)
* How are the clips determined? Based on shot boundary detection from Panda-70M
* What is N? Adaptive? How many levels are there, and do the thoughts vary significantly?

**Comparison to Single-Cue SFT**: In the ablation study for tree-guided visual cue localization (Table 4), the "Single-Cue-SFT" is introduced. Could you elaborate on how the rationales in the Video-SingleCue-SFT-1k dataset were generated? Was it simply by prompting the LLM with only the final key clips?
* I’m surprised by how well Tree-of-Cue performs over Single-Cue in Table 4, especially given the minor improvements from the Baseline to Single-Cue on MMVU and VideoMME (overall the additions on VideoMME are weaker). It would be good to understand why the behavior is so different between MMVU and MVBench. I suspect general descriptions of the video help the model, not just tree-guided cues.
* A nice ablation would be to compare randomly selected time segments (containing the leafs still), to see if it’s simply the additional information present which improves the model’s video content analyzing capabilities.

**Distributed Visual Cues**: How would the Video-ToC framework handle videos where the crucial information for answering a question is not concentrated in a few key clips but is rather distributed across the entire video or requires understanding the evolution of a scene over a longer duration?

---

### Note · Authors · 2025-11-12

I have read and agree with the venue's withdrawal policy on behalf of myself and my co-authors.